# Towards Sustainable and Smart Cities: Replicable and KPI-Driven Evaluation Framework

Ana Quijano [1],*, Jose L. Hernández [1],*, Pierre Nouaille [2],*, Mikko Virtanen [3],*, Beatriz Sánchez-Sarachu [4],*, Francesc Pardo-Bosch [5],[6],* and Jörg Knieilng [7],*

1   Foundation CARTIF, Parque Tecnológico Boecillo 205, 47151 Boecillo, Spain
2   CEREMA, Direction Territoriale Ouest, MAN, Rue René Viviani, 44000 Nantes, France
3   VTT Technical Research Centre of Finland Ltd., P.O. Box 1000, 02044 Espoo, Finland
4   TECNALIA, Parque Científico y Tecnológico de Bizkaia 700, 48160 Derio, Spain
5   Center for Public Governance (ESADEgov), Universitat Ramon Llull-ESADE, Pedralbes, 60, 62, 08034 Barcelona, Spain
6   Department of Project and Construction Engineering, Universitat Politècnica de Catalunya (BarcelonaTech), Jordi Girona 1-3, 08034 Barcelona, Spain
7   Department of Urban Planning, HafenCity University Hamburg, Henning-Voscherau-Platz 1, 20457 Hamburg, Germany
*   Correspondence: anaqui@cartif.es (A.Q.); josher@cartif.es (J.L.H.); pierre.nouaille@cerema.fr (P.N.); mikko.virtanen@vtt.fi (M.V.); beatriz.sanchez@tecnalia.com (B.S.-S.); francesc.pardo@upc.edu (F.P.-B.); joerg.knieling@hcu-hamburg.de (J.K.)

**Abstract:** Sustainability is pivotal in the urban transformation strategy in order to reach more resource-efficient, resilient and smarter cities. The goal of being a sustainable city should drive the decisions for city interventions, and measuring city progress is a key step for this process. There are many initiatives aiming at defining indicators and assessment procedures, but there is no convergence in the definition of terms and application methodologies, making their real implementation complex. Within mySMARTLife project (GA#731297), a KPI-driven evaluation framework has been defined with the aim of covering the multiple pillars of a smart and sustainable city (i.e., environment, energy, mobility, ICT, citizens, economy, governance) in a holistic way. This methodology also defines the concepts and terms to guide urban planners and/or experts at the time of implementing the framework for any specific city. The evaluation framework has been deployed in the cities of Nantes, Hamburg and Helsinki, and some lessons have been learned, such as the necessity of providing a definition of measurement boundary to avoid biased interpretations. Due to a co-creation strategy, the main issues from the cities have been taken into consideration in order to increase the replicability of the results.

**Keywords:** sustainable cities; evaluation framework; indicator; smart city; energy efficiency; renewable solutions; electromobility

## 1. Introduction

Cities have been transformed into hubs for modern civilizations [1], but this transformation has impacts in the use of the natural resources. The limited nature of these resources motivates the necessity for more sustainable cities, which may be achieved by the application of new technologies [2]. In this direction, the European Commission has adopted an ambitious plan for the decarbonisation of European cities and the penetration of renewable energy sources. European targets consist of a reduction in GHG emissions by 55% in 2030 and climate neutrality in 2050 [3].

The population prediction models indicate that more than 85% of European citizens will live in urban areas by 2050 [4]. Smart energy, sustainable mobility, smart people and economy then become the key topics for urban transformation [5]. All of them are

supported by the ICTs (Information and Communication Technologies) as the enabler for digital cities.

mySMARTLife project (grant agreement #731297) [6], which is H2020-funded, aims at the sustainable and smart transformation of the three lighthouse cities of Nantes, Hamburg and Helsinki by applying the aforementioned concepts. More than 150 actions are contributing to the pillars of energy, mobility, ICT, citizens, economy and governance to the new urban sustainability concepts. mySMARTLife targets are: (a) renewable share of 54%; (b) reduction of 55% of greenhouse emissions due to buildings, city infrastructure and mobility actions. Furthermore, mySMARTLife fosters both smart economy and smart people supported by local economy growth and entrepreneurship. The main driver for designing and implementing this innovative concept of smart city is integrated urban planning.

Nevertheless, the measurement of the impacts achieved requires a rigorous assessment plan with a focus on the quantification of the final numbers in terms of sustainable transformation. This paper presents the evaluation framework that has been prepared and deployed within mySMARTLife project. This framework aims at merging the multiple and diverse verticals of the city with the objective of reaching a global view of the smart city. A set of KPIs (key performance indicators) allows for the implementation of the plan in order to obtain the final impacts with an objective procedure. A four-step methodology has been followed to reach this result, being the fourth step covered in this paper, with the aim of the definition of an affordable and objective framework that evaluates the sustainability of cities across multiple city verticals.

The paper is structured as follows: Section 2 provides a set of references and previous research in the implementation of urban sustainable evaluation plans. Section 3 presents the methodology that has been applied in mySMARTLife project for the definition of the indicators and the evaluation framework. Next, Section 4 describes the framework and how it has been applied in the different pillars across cities. Finally, Section 5 compiles the set of conclusions and lessons learned.

## 2. Background: Other Sustainable Evaluation Frameworks

Recent studies have tried to provide an answer to the quantification of the impacts that a set of actions could have in a smart city. There is a big diversity of methods focused on the measurement of the sustainability and smartness, but the complexity and multidimensionality of these concepts are one of the major existing barriers [7]. There are standards for sustainable ratings of the built environment (e.g., LEED [8], BREEAM [9] or CASBEE [10]), which have been adapted to city context [11]. However, these frameworks prioritize environmental sustainability over economic and social sustainability [12]. In addition, the way of application of these methodologies is based on the assessment of multiple criteria by comparison to benchmarking values, which restricts their use only to countries that have already defined those baseline values [13].

Standards such as ISO 37120 [14] or ISO 37122 [15], as well as the indicators for Sustainable Development Goals for the implementation of the 2030 Agenda for Sustainable Development [3], should be the reference for the evaluation of smart cities under a sustainable approach. Nevertheless, although they are good references, the difficulties fall in their application. While indicators are well defined, there are no methodologies to guide cities and experts in the time of deploying such evaluation frameworks.

The European Commission has currently funded 18 projects for promoting European smart cities and communities towards the energy transition. Complementarily, the initiatives CITYkeys [16] and smart cities marketplace [17] support these projects through the definition of a set of indicators applicable in multiple axes (e.g., planet, people or prosperity, among others), having the main goal of being able to compare the European cities and rank them. Many of these EC-funded projects make use of the indicators defined by CITYkeys, but how the assessment is applied differs from project to project, even when similar smart city solutions have been implemented.

Angelakoglou et at. [18] propose a categorization of KPIs in six dimensions (technical, environmental, economic, social, ICT and legal), and the performance metrics to measure each KPI is selected by each demonstrator city among these options: baseline, business as usual or another different threshold. Kourtzanidis et al. [19] summarize several insights on key characteristics and limitations of currently available urban sustainability and smartness evaluation frameworks and propose indices in a smart city evaluation framework under a triple axis approach: the project performance index, the sustainability impact index and the sustainable performance index. Finally, García-Fuentes et al. [20] define a framework based on energy and mobility, while the social and economic aspects are in the background. Moreover, the evaluation plan relies on an index that normalises the sustainable rank of the city by weighing the actions and prioritising energy and mobility.

Consequently, there is a not a common and accepted evaluation framework for city assessment. Different methods are defined in each of these initiatives, which differ on the type of metrics (indicators/indexes as rating systems), the scope of the evaluation (only focused on the intervention area/extrapolation at city level) and the objectives and domains to be measured. These issues limit the application of the evaluation frameworks in a real context; meanwhile, many of them are used for scientific purposes. mySMARTLife evaluation framework, presented in this paper, goes a step forward by two main aspects: (1) it takes advantage of the previous assessment plans analysed in the previous research projects to exploit the benefits and lessons learned from them; (2) it defines a clear and affordable method for its implementation across cities. In this sense, first, the framework overcomes both the intervention area level and the city extrapolation. Second, it is not restricted per domain (i.e., covers both energy and mobility and other pillars at the same level), which provides more flexibility and adaptability to the city actions (e.g., evaluating the performance of the application of a directive for the promotion of renewable energy). Third, the framework is defined using a co-creation strategy with the city experts and urban planners to overcome real problems at the same time of deploying assessment frameworks, allowing for better replicability.

## 3. Materials and Methods

The development of an evaluation framework is a key requirement for the calculation of impacts since this sets up the purpose to evaluate and defines the elements to be used for the evaluation. The framework that was defined in mySMARTLife is driven by KPIs with a two-fold scope: city and project (intervention area) levels. Multi-dimensions have also been considered to cover the multiple pillars in a city: energy/environment, mobility, urban/ICT infrastructure, citizens, economy and governance.

The design of this evaluation framework has been the result of the collaboration among research centres, technology providers, and cities, thus creating a co-creation strategy between stakeholders. This strategy allows for a holistic perspective that places the sustainable goals of the cities into value. To support such development, a four-step procedure has been followed, as described below, although the main focus of this paper is on the fourth step, which defines and deploys the evaluation framework.

(1) Step 1 establishes the objectives of the evaluation through the definition of relationships among the foreseen impacts of the actions and the cities vision compiled in their city plans. During this step, the technical, environment, social and economic expected improvements within the demonstration actions are analysed. In addition, it includes the focus and targets of the city models on the thematic areas of climate change, energy, mobility and ICT within the short-term plans (2020–2030). Additionally, the concepts "smart people" and "smart economy" developed in mySMARTLife are addressed. Thus, the evaluation framework is set to measure the progress obtained in the cities under this approach: (1) to achieve an economic growth decoupled from resource use to face the current pollution and $CO_2$ emissions; (2) to improve the life quality of citizens; and (3) cities operate in a more efficient way. As result of this step, the

"human" language objectives are obtained, which need to be mapped into dimensions and quantifiable indicators included in steps 2 and 3 of the methodology.

(2) Step 2 identifies the dimensions to evaluate the evaluation framework. These dimensions are extracted from the well-established ones from CITYKeys [16], complemented with experiences from mySMARTLife on how to allocate the impacts to be reached to the project technologies. Initially, a document is defined by the technical partners involved in the project, which are updated and validated by city partners during diverse iterations and dedicated workshops. The output of this work is shown in Table 1 below, which is depicted to understand the next steps of the methodology.

(3) Step 3 maps the available indicators to the objectives and dimensions defined within steps 1 and 2. mySMARTLife made the exercise of collecting the list of indicators from literature and reference projects, as summarized in Table 2 and as available in [21], in order to create a dataset of indicators to be applied to the demonstrated solutions. From this list, the suitable indicators for each dimension are selected so as to quantify the final impacts and results of the actions, i.e., each city or project does not need to define its own indicators, but needs to choose the appropriate ones from the list.

**Table 1.** Dimensions for evaluating the different smart city visions with the solutions to be applied and the expected impacts and performance.

| Smart City Vision | Dimensions to Evaluate | Solutions | Expected Impacts and Performance of Solutions |
| --- | --- | --- | --- |
| Sustainable use of resources/Quality of life | Energy and Environment | Efficient Buildings/District and City infrastructure | Reduction in energy consumption Decrease in GHG emissions RES production Energy delivery in the system Fraction of energetic self-supply by RES |
| | Mobility | Clean vehicles | Decrease in GHG emissions, $NO_X$ and PM emissions Amount of use Energy consumption Safety |
| | | Charging stations and solar road | Use and usage pattern Energy demand management Degree of energy supplied to EV by RES |
| | | Last mile delivery and multimodality | Willingness to invest/use |
| City operational efficiency | ICT and Urban platform | Urban platform and ICT developments | Impact in digital transformation Performance of ICT services |
| Prosperity/Quality of life | Economy | Innovative business | Monetary impacts of the demonstrative actions in the cities, citizens and companies Cost-effectiveness |
| Community involvement | Citizens | Citizen engagement | Social acceptance on project solutions Citizens reached in citizens engagement activities |
| Sustainable resources/Quality of life/Prosperity/Efficiency/Citizens involvement | Governance | Urban planning, policy improvements and staff exchange | Impact of the project in the city urban planning and policy improvement |

**Table 2.** Relevant indicator references used for the definition of the indicators.

| Evaluation Framework | Literature | Reference Projects |
| --- | --- | --- |
| City level framework | Agenda for Sustainable development of the United Nations, standards ISO 37120 and ISO 37122, Eurostat City Statistics, Covenant of Mayors, CITYkeys, SCIS and United for Smart Sustainable Cities (U4SSC) | SmartEnCity, REMOURBAN, Replicate and CITyFiED |
| Project level framework | CITYkeys, SCIS, Eurbanlab, World Bank, OECD and Telefónica Foundation | SmartEnCity, REMOURBAN, Replicate and CITyFiED |

As a result of this process, 151 city level indicators and 128 project level indicators have been defined in an iterative process among technicians from research centres and city partners following the following criteria:

- Measurability: The identified indicators should be capable of being measured through the data collection methods established in the project;
- Completeness: The indicators should cover all the type of interventions (district, city infrastructure, mobility, ICT) and non-technical aspects (governance, citizens, finance) deployed in the project as well as the expected type of impacts (environment, economy, social and technical);
- Relevance of the indicator for the purpose of the evaluation defined;
- Availability of data in the cities for the final selection of indicators since not all relevant indicators can be quantifiable.

Conversely, to guarantee a proper evaluation of impacts, some considerations have been also included below:

- Independence and non-redundancy of indicators;
- Familiarity of the persons in charge of evaluation with indicators, through a good description of the formulas and definitions and dedicated sessions to clarify possible doubts.

(4) Step 4, which is the main purpose of this paper, defines the KPI-driven evaluation framework, combining the objectives and indicators for the project domains. This step also includes the specific assessment plans to evaluate the impacts of the interventions, based on the selected KPIs. This is the main result of this paper, which is explained through the next section.

## 4. mySMARTLife KPI-Driven Evaluation Framework

As stated before, the result of the methodology is the assessment framework, which has been applied in the cities of Nantes, Hamburg and Helsinki.

Figure 1 [21,22] illustrates the proposed framework. First, the two levels of assessment that are included should be highlighted:

- Project level includes more than 150 actions that are being deployed in the specific areas of the cities involved in the project. Then, the main objective is to obtain the quantitative analysis of the impacts achieved after those actions (e.g., building retrofitting, integration of renewables, electrification of the transport, etc.) as well as the performance of the technological solutions;
- City level, which extrapolates the quantified impacts from the project, to estimate the impact that these actions would have in the city. The outcome of this level is the support of the cities, and at the same time the planning of urban transformation strategies by following quantitative and objective methods driven by KPIs.

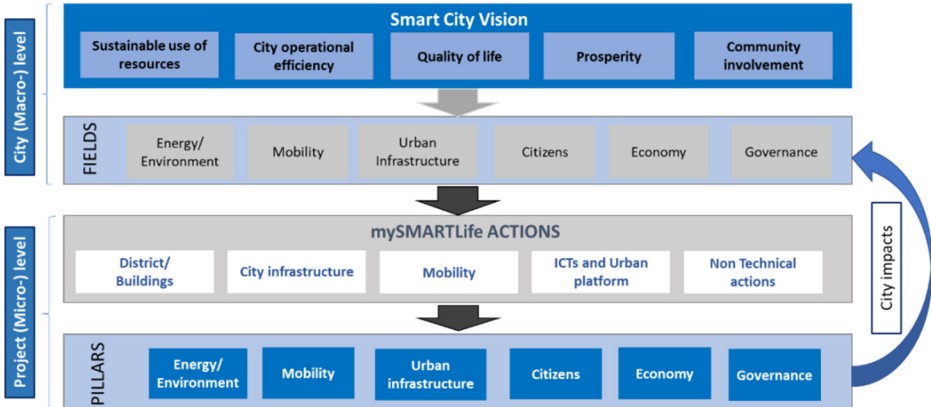

**Figure 1.** mySMARTLife KPI-evaluation framework.

Within each of the stated levels, a set of categories (named fields in the city level and pillars at the project level) are established. These are, as mentioned beforehand, energy and environment, mobility, urban infrastructures (including the digitalisation of the city through the ICTs and urban data platforms deployment), citizens, economy and governance. All of them are driven by a set of indicators [21,22], which are summarised in Table 3.

**Table 3.** Number of indicators defined per pillar.

| Core Categories | City Level | Project Level |
|---|---|---|
| Energy and Environment | 56 | 32 |
| Mobility | 22 | 51 |
| Urban infrastructure (digitalization by ICTs) | 20 | 11 |
| Economy | 16 | 22 |
| Citizens/Social | 16 | 5 |
| Governance | 15 | 7 |

City level indicators are calculated with the data compiled from public databases, mainly from city statistics. For the case of project level indicators, two KPI categories have been established:

- Quantitative indicators to demonstrate the impacts of innovative solutions through the collection of data from meters and other data compilation methods;
- Qualitative indicators to assess the perception of benefits gained by citizens, companies and the municipality through questionnaires and surveys.

Finally, the framework complements the indicators and the definition by methodologies and protocols with the goal of supporting cities during the implementation of the project evaluation framework. Thus, the evaluation framework not only provides a theoretical indicator-based procedure, but also pathways to apply them to analyse the success of the implemented actions. Table 4 [22] depicts a summary of the proposed methodologies for each of the project pillars.

**Table 4.** Evaluation methods for each one of the categories.

| Core Categories | Evaluation Methodology |
|---|---|
| Energy and Environment | Extension of IPVMP |
| Mobility | $CO_2$ emissions-based |
| Urban infrastructure (digitalization by ICTs) | Software metrics |
| Economy | Cost-Benefit |
| Citizens | Surveys |
| Governance | Questionnaires |

### 4.1. Analysis of the Project Pillars and Categories

The general framework is applicable in multiple verticals of the city. As indicated in Table 4, although the framework is defined in a holistic way to consider the cross-domain effects, each of the categories requires its specific evaluation methodologies. These are described in the next subsections.

#### 4.1.1. Energy and Environment

Energy and environment pillars refer mainly to energy efficiency in the built environment and other elements of the cities that imply energy consumption reduction such as smart lighting or renewable generation at the local and district/city levels (e.g., building-integrated RES, district heating, PV plants, wind farms, etc.).

To determine high-performance districts, the energy demand and the use and the self-consumption of the buildings are calculated. In order to accomplish this, IPMVP (International Performance Measurement and Verification Protocol) [23] has been selected, as it is a standard for the evaluation of the energy performance. Explaining IPMVP is

not the objective of this paper, but how it is adapted to the mySMARTLife framework requirements. In this sense, two measurement periods are established:

- Baseline: This represents the starting point, i.e., the reference for comparison. Three methods are available:
  - i. Using the country normative for new and/or existing buildings as reference;
  - ii. Simulate the energy behaviour of the building through any simulation software (also applicable for new and/or existing buildings);
  - iii. Only for existing buildings and in case of monitoring is available, energy performance based on real data (smart meters or energy bills) is calculated.

- Reporting period: It is the period after the construction or renovation of the building, where the final performance is measured. This period has the requirement of real data, either monitored with smart meters or obtained from energy bills.

These two periods are then compared to obtain the final impact, but it needs adjustments, such as climate conditions. This is a routine procedure well established by IPMVP. However, what is even more important is the definition of the boundary, which is one of the main lessons learned, as described in Section 4.2. Figure 2 [21] shows how mySMARTLife defines the different boundaries to create a common understanding when the evaluation procedure is applied. Many of the existing frameworks fail in the definition of the boundary, consequently generating confusion and complexity, and this is how mySMARTLife solves it.

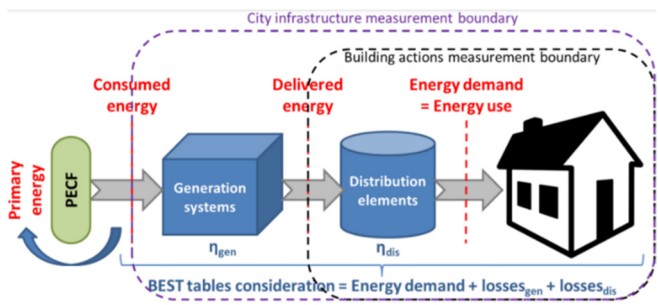

**Figure 2.** Assessment boundary for the energy and urban infrastructure categories.

Two levels are established in this pillar: building actions and city infrastructure. mySMARTLife sets the boundary for building actions as the combination of the energy demand or use together the delivered energy. As depicted, the boundary surrounds all the elements of the building, including the local renewable energy or local generation systems (e.g., individual boiler) that are used for self-consumption.

When applying at district/city level for shared generation systems (e.g., district heating), the boundary is rescaled (as drawn in Figure 2). It does not only contain buildings and distribution elements, but also integrates such generation systems to calculate the indicators at consumed energy levels (i.e., considering the performance of the different elements in the generation and distribution chain).

Finally, the case of lighting systems comprises the energy consumption of the bulbs and the comparative of energy when light bulbs have been replaced. In this specific case, the adjustment is not made based on climate conditions, but on hours of use.

### 4.1.2. Mobility

The mobility evaluation pursued the quantification of mobility actions impacts and performance in terms of:

- Reduction of air quality emissions due to replacement of ICE (internal combustion engine) by electricity powered vehicles (EVs);
- Amount of travel, energy consumption and journey quality of e-vehicles;
- Amount of use and pattern of the charging infrastructure installed;

- Degree of energy managed and supplied to EVs by renewable sources;
- Willingness to use multimodality actions and investment in urban freight.

Data collected from transport facilities are used for the calculation of KPIs, identified by each mobility action with the exception of the impacts in the air quality emissions that need of a specific methodology to quantify the avoided air emissions. Thus, the evaluation approach in mySMARTLife establishes two measurement periods: baseline with ICE vehicles as reference for comparison and reporting period with e-vehicles. Additionally, this considers that distances travelled during both periods are the same. Then, the emissions avoided are measured as a function of consumed fuel or distance travelled per each type of vehicle and applying different emissions factors to each energy source used by them (e.g., diesel, electricity, etc). Standard emission factors for fuels are provided for European countries by the Covenant of Mayor and internationally by IPCC, whereas average consumption per distance travelled for each vehicle is shared by its manufacturer.

This means that the vehicle features (energy consumption and type of fuel consumed) are the only factors that change among baseline and reporting period, whereas other external factors to the vehicle are not analysed since the interventions do not have any influence over them (e.g., driving speed, driving style, road characteristics, traffic and weather conditions).

### 4.1.3. Urban Infrastructure/Digitalization via ICTs

Digitalization of the city is also considered in this evaluation framework, which is reached through the implementation of ICT solutions in form of urban data platform. The method for the ICT analysis diverges from the previous infrastructure analysis, as the domain is completely different. In this specific case, software metrics are used to measure the level of digitalization of the city. Basically, the ICTs are quantified as:

- Number of sensors and datasets integrated in the urban platform;
- Number of available services;
- Number of available open data and open APIs (application programming interface);
- Number of different users, such that usability can be determined;
- Response time, as performance metric to determine the time that any user should wait to receive the expected results from the urban platform services;
- Scalability, as the capability for extending the resources of the urban platform;
- Availability, as the time during which the urban platform does not suffer crashes.

### 4.1.4. Economy

The economy pillar has as objectives the measurement of the actions' effectiveness and the related business models, as well as the monetary impacts of the demonstrative actions in the cities, citizens and companies involved in their implementation. An analysis of cost–benefit of the solutions is performed after the calculation of KPIs identified with the data provided by the action leaders once the actions are concluded.

The economic evaluation is then implemented as follows:

- Financial performance of the actions through the description of the funding/financial model and the identification of the costs and revenue structure associated with the implementation, operation and maintenance of the actions;
- Societal, economic and environmental benefits of actions in terms of monetary terms through the evaluation of a variety of aspects such as: jobs created, expenditure in local economy, impact in business units and improvement in air environmental quality among others.

### 4.1.5. Citizens (Social)

This pillar tries to reveal the degree of satisfaction of citizens with the project solutions deployed in the city and analyse the existence of a behavioural change in the society as well as the factors that influence in the level of acceptance. The analysis is rendered

through tailored questionnaires according to the object to be assessed and the target audience defined which must be the citizens affected by the interventions. The tool, which is distributed once the project actions have finished, allows for the evaluation of the final acceptance of the local population about new technologies installed, the willingness to invest in similar solutions and/or recommending these to others. This analysis also includes citizens' perception in the technical and economic design of the solution, the amount of information received and the degree of involvement in decision making. Finally, an analysis of the respondent profile is performed (e.g., age, gender, socio-economic status) for considering this result in future social campaigns focused to upscale/replicate the solutions evaluated.

Additionally, this pillar is addressed to assess the target people reached in citizen engagement activities carried out by the project to inform about benefits of energy efficiency and RES solutions and to empower citizens in the urban transformation planning process. To this regard, the number of people reached and the diverse social background are evaluated.

4.1.6. Governance

The governance pillar aims to assess how the project has contributed to the urban development by the means of a questionnaire based on Likert Scale and open questions, which is filled in by the main contact point of each lighthouse city at the end of the project. Main aspects to be gathered correspond to:

- Function of the local authority in the development of the project: role in the financing, implementation, management and transferability of knowledge gained;
- The extent to which the project has been able to influence in the local government with re-definition of city policies and the implementation of changes in the organizational scheme of the local administration or development of new rules and regulations;
- To which extent the project has influenced in the identification of city priorities and most promising solutions to achieve the city vision:
  - ○ How Sustainable Energy and Climate Action Plans (SECAP), Sustainable Urban Mobility Plan (SUMP) and others city plans have been benefited from the lessons learned during the implementation of actions;
  - ○ How methods applied during the definition of an innovative urban transformation strategy and the outputs obtained from energy demand of the cities, energy scenarios, techno-economic analysis and business models have contributed to the definition of a long-term advanced planning in the city.

*4.2. Implementation in the Cities of Nantes, Hamburg and Helsinki: Discussion and Lessons Learned*

This framework is, as mentioned before, deployed in the cities of Nantes, Hamburg and Helsinki. During the implementation, the main results are translated into relevant lessons learned that have been finally collected. At the design level of the evaluation framework, there is a set of challenges that needs to be considered:

- Co-creation strategies are crucial for a successful application. Within the definition of the objectives to evaluate (step 1 of the methodology), different stakeholders of the smart city should be involved, counting on the experience of experts in different disciplines (energy, mobility, ICT, social, economy and governance).
- Aligned with the co-creation, the definition of indicators (steps 1 to 3) is a joint process, including external partners such as research centres and universities, taking advantage of the experiences and the alignment with the state of the art. Due to this process, a complete, holistic and comparable evaluation framework may be deployed to guide city stakeholders in the whole decision-making process.
- The design of an evaluation framework (step 4) is a live process. Unforeseen events usually arise during the implementation of actions, which lead to an update of the interventions to be deployed.

- The definition of the objectives (step 1) requires knowing the whole context of the action to be executed and their expected impacts (dimensions in step 2) in the city as a key requirement for searching for suitable indicators (step 3).
- The evaluation of innovative aspects in demonstrative solutions can require the definition of new indicators or adaptations of the formulas to the case of study (step 3). This process requires of certain experience and is usually more complex than initially planned.

During the deployment of evaluation framework, a set of conclusions and lessons learned are also extracted.

- At the time of applying the framework, the definition of evaluation boundaries of the action under evaluation is the key for a common understanding of what to evaluate.
- High quality of data are not reached in all the timelines in which an action is under evaluation. Non-completeness of data and outliers are common issues that appear during the data collection.
- A follow-up process of the data collected through meters is needed to identify the best period to calculate KPIs and to assure that the KPI calculation has been made with enough quality data, therefore, being able to certifying the impact calculation accuracy.
- The coordination among different evaluation processes is the key to guarantee the proper evaluation from a holistic approach. Two main figures should be considered for the evaluation of the actions: a main person responsible of all the evaluations and a main evaluation contact person for each city involved. Additionally, despite the proper coordination for reporting quality data and KPIs, different templates are used for different actors in the evaluation, which involve a more complex process than initially expected.
- The selection of standard protocols and methodologies in the various verticals (e.g., IMPVP for energy) generate a high degree of confidence and accuracy for the analysis of the real achievements and/or impacts. The framework presented in the paper is flexible enough to set up ad hoc procedures to calculate baseline and reporting periods in order to make them. This is not a trivial task, and it is neglected many times, generating non-realistic or distorted view of the impacts due to wrong assumptions.

## 5. Conclusions

This paper has presented a KPI-driven evaluation framework defined in mySMARTLife project that allows cities to measure the achieved impacts through the means of the deployment of sustainable actions as well as the progress achieved towards the compliance of city targets established in energy transition urban plans.

mySMARTLife evaluation framework consists of a broad, flexible and replicable methodology that has been established to guide decision makers in how to face the main challenges of the analysis, i.e., quantification of the results or determining the main goals to evaluate. This relies on a list of available indicators (merging literature and previous experiences) that helps cities to select the most suitable ones according to the objectives to be reached. Cities are thus capable of mapping the KPIs with the expected targets or smart city urban plans: not only by choosing them from the pre-defined list, but by also adapting the indicators or the components of the evaluation framework to their requirements.

This framework has been validated in the lighthouse cities of Nantes, Hamburg and Helsinki as tool to assess the effects of the more than 150 actions implemented into the environment, economy, citizens and urban planning contexts. In this case, the evaluation is not only focused in technical and environmental aspects but also in finding non-technical barriers, such as low social acceptance and low profitability of technologies or the necessity of policy improvements. Consequently, valuable insight can be obtained from the evaluation performed with this framework that can lead to establish measures to boost the scale up of the sustainable solutions in the cities.

The flexibility of the presented evaluation framework allows for its application in the required context. The presented framework is iterative, being able to reassess the objectives

and actions, therefore supporting decision makers and softening its complexity. Moreover, the replicability feature of the framework allows other cities to look at previous experiences. However, the main limitation of the study is the limited number of cities addressed until now. Nantes, Hamburg and Helsinki are partners of the mySMARTLife project with a clear definition of actions to be deployed across the city. In the case that other cities would be considered, they could require a previous step to identify the real challenges.

At the time of writing this paper, the mySMARTLife project is delving deep into the monitoring stage to calculate the KPIs defined, which will drive the final assessment. The final results will be obtained in the upcoming months to extract conclusions and to run a new iteration in the cities. This is the nearest future line for this research, while other future lines are focused on expanding the framework: first, the unforeseen situation of COVID-19 and the implications in the smart city plans in order to adapt the framework to such un-predictable situation; and second, other verticals will be researched to be integrated into the framework (e.g., waste management).

**Author Contributions:** A.Q. and J.L.H. have been the main contributors and authors in the conceptualization of the general assessment framework and evaluation methodology. The definition of the KPIs has been a collaborative work guided by A.Q., where J.L.H., P.N., M.V., and B.S.-S. have focused on the energy pillar. A.Q. and P.N. have defined the mobility part, and J.L.H. has defined the ICT and urban platform pillar. F.P.-B. was responsible for the economic vertical, J.K. for social aspects, and B.S.-S. for governance, in all the cases supported by A.Q. Finally, P.N. has been in charge of the deployment and validation of the framework in Nantes, including with M.V. in the case of Helsinki. All authors have read and agreed to the published version of the manuscript.

**Funding:** The work presented in this paper is the result of the EU-funded project mySMARTLife, under the H2020 programme with grant agreement no. 731297.

**Institutional Review Board Statement:** Non applicable. The study does not involve humans or animals.

**Acknowledgments:** The authors would like to thank, first, the EC for funding and supporting mySMARTLife project under GA#731297. Second, the authors would like to thank the rest of consortium of mySMARTLife for the contributions in the definition of the framework.

**Conflicts of Interest:** The authors declare no conflict of interest.

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
