# Peer review of "Towards Sustainable and Smart Cities: Replicable and KPI-Driven Evaluation Framework"

_buildings, doi:10.3390/buildings12020233_

Round 1
Reviewer 1 Report
- Authors are to undertake a professional proofread of the manuscript for grammar and syntax errors.
- The manuscript will benefit more from a separation of the Discussion and Conclusion section. A Discussion section will give more clarity to the results from the evaluations, and also demonstrate better the implications of the study to theory and practice.
- The revised Conclusion section should start by re-stating the aim of the study and then a summary of the findings. It will be beneficial if the authors could provide more suggestions on the adoption/implementation of the mySMARTLife evaluation framework. Following this up with suggestions for future works will benefit the paper. It is obvious that there are limitations in the current study (method/approach, research coverage, etc.) which authors could suggest for future research investigations.
Reviewer 2 Report
The whole manuscript requires a careful proof read. The grammatical errors and typos need to be fixed.
i.e. line 111) iniciatives?
Use of commas and semicolons need to be revised, this has made
some sentences are difficult to follow:
i.e line 114 and 115: “These issues make even more complex its real deployment in cities.”
The papers organisation, logic and connections are poor at this stage and require major amendments.
The methodology does not explain the method of analysis; for instance, how are the objectives established in step one? Where is the results of this step discussed its difficult to identify and follow?
Accordingly, how is the identification of dimensions done in step 2. And, Selection of the suitable indicators.
The discussion should be cross-referenced with the results of each step mentioned in the methodology.
The preference is to separate discussion and conclusion. The discussion has to be mainly about the results obtained. In the conclusion section generally, new material should not be included, and mainly the theoretical and practical impact of the work should be included. How is this framework going to be applied our further developed to enhance the goal of Sustainable and Smart Cities.
The limitations of the work also need to be explained.
Round 2
Reviewer 2 Report
Not much has been changed, the results of step one are still difficult to follow. step one states "Establishing the objectives pursued by cities involved in mySMARTLife de-138 fined in their urban plans to setting the basis of the evaluation."
The question is have the authors established these objectives and if so where in the text can we find the process of establishment? and if city plans were analysed for this what kind of analysis was use? still not clear!
For step 2 method of analysis and identification of dimensions should be explained, was it document review, content analysis, interviews, etc.
It is best to completely separate the methods from the results so its less confusing to the reader.
for step 3 the reader cannot establish a link between the objectives mentioned in step one and the indicators. I suggest its best that the authors organize the indicators based on the objectives so the logical link can be visible. You may have used the words "criteria" and objectives interchangeably which makes it quite confusing.
Looking at section 4.2 and conclusion it seems that not much has changed from the original version.
